# Establishment of a Real-Time Recombinase Polymerase Amplification for Rapid Detection of Pathogenic *Yersinia enterocolitica*

**DOI:** 10.3390/pathogens12020226

**Published:** 2023-01-31

**Authors:** Hongjian Zhang, Meng Zhao, Siyun Hu, Kairu Ma, Jixu Li, Jing Zhao, Xin Wei, Lina Tong, Shengqiang Li

**Affiliations:** College of Agriculture and Animal Husbandry, Qinghai University, Xi′ning 810016, China

**Keywords:** *Yersinia enterocolitica*, recombinase polymerase amplification, *ail* gene, rapid detection

## Abstract

*Yersinia enterocolitica* is a zoonotic proto-microbe that is widespread throughout the world, causes self-limiting diseases in humans or animals and even leads to sepsis and death in patients with severe cases. In this study, a real-time recombinase polymerase amplification (RPA) assay for pathogenic *Y. enterocolitica* was established based on the *ail* gene. The results showed that the RPA detection for *Y. enterocolitica* could be completed within 20 min at an isothermal temperature of 38 °C by optimizing the conditions in the primers and Exo probe. Moreover, the sensitivity of the current RT-RPA was 10^−4^ ng/μL, and the study found that the assay was negative in the application of the genomic DNA of other pathogens. These suggest the establishment of a rapid and sensitive real-time RPA method for the detection of pathogenic *Y. enterocolitica*, which can provide new understandings for the early diagnosis of the pathogens.

## 1. Introduction

*Yersinia enterocolitica*, a member of the *Enterobacteriaceae* family, is a kind of zoonotic pathogen that exists widely all over the world [1,2]. It can infect humans or animals through contaminated food or water, causing self-limiting disease, lymphadenitis and terminal ileal inflammation, sepsis and even death in severe cases [3,4].

*Y. enterocolitica* can be classified into six biotypes: 1A, 1B, 2~5, and more than 70 different serotypes [5]. All biotypes, except for type 1A, are pathogenic to varying degrees in any animal. Because biotype 1A does not have the virulence plasmid pYV and chromosomal virulence gene, it is usually considered to be non-pathogenic [6,7]. The major virulence genes of pathogenic *Y. enterocolitica* are distributed in *ail*, *fyuA*, *ystA*, *virF* and *yadA*. While the plasmid virulence genes virF and yadA are easily lost during passage, the ail gene is an important virulence marker of Yersinia enterocolitica and is widely used in pathogenicity analysis [8].

Yersinia, caused by the bacterium *Y. enterocolitica*, is now the third most common zoonosis in Europe after campylobacteriosis and salmonellosis [9]. Previous reports showed a 166% increase in Yersinia patients in 2017 compared to 2014–2016 [10]. The detection rate of *Y. enterocolitica* in common food was 2.33% and in frozen food was 6.72% [11]. At present, *Y. enterocolitica* has become an important pathogen detected in many countries.

The traditional isolation method of *Y. enterocolitica* requires a lot of time, and most isolates are non-pathogenic strains [12]. The molecular methods of the PCR, multiplex PCR, real-time PCR and loop-mediated isothermal amplification (LAMP) have been reported for the diagnosis of pathogenic *Y. enterocolitica* [8,13,14,15,16]. The molecular detection method is able to detect *Y. enterocolitica* in a shorter period of time compared to the traditional method, which takes a lot of time. Since the introduction of the classic PCR in 1983, nucleic acid amplification–based pathogen detection has been widely used for disease surveillance [17]. However, it has only been used in laboratories because it requires sophisticated thermal cycling apparatuses and trained personnel. Although the PCR and real-time PCR display high sensitivity, accuracy and specificity, these methods require specialized equipment and experienced operators, thus limiting their application [18]. The LAMP assay for *Y. enterocolitica* allows for more sensitive analysis than does PCR detection, although it requires the design of multiple primer pairs and takes more time [15]. Therefore, a simpler and faster test is necessary for the quick diagnosis of pathogenic *Y. enterocolitica*.

Many strategies to achieve amplification at a constant temperature have emerged as potential alternatives to PCR, with the recombinase polymerase amplification (RPA) being the fastest growing in recent years [19]. The RPA assay is a highly specific and sensitive detection technique compared with other existing detection techniques, requiring only 10–20 minutes to complete the reaction with only a primer of 30–35 bp to complete the amplification between 25–42 ℃ [16,20,21]. Therefore, the RPA replaces the thermal cycling required for PCR by using recombinant enzyme proteins, single-stranded DNA binding proteins for primer annealing and a strand-swapping DNA polymerase for amplifying nucleic acid sequences [22]. The RPA reaction has been widely used in the detection of various pathogens due to its advantages of simple operation, strong specificity, high sensitivity and fast detection speed [21,23,24,25]. Real-time RPA has been successfully used to detect a variety of pathogens, including *Vibrio vulnificus* [26], *Avian reovirus* [27], and *Schistosoma haematobium* [22]. Hence, this study focused on the development and validation of a novel RT-RPA assay for the simple and effective detection of pathogenic *Y. enterocolitica*.

## 2. Materials and Methods

### 2.1. Bacterial Strains and Culture Condition

Seventeen bacterial strains were applied in this study, including the five pathogenic *Y. enterocolitica*, five non-pathogenic *Y. enterocolitica* and seven non-*Y. enterocolitica* bacterial species (Table 1). The pathogenic *Y. enterocolitica* ATCC 23715 was obtained from the China Institute of Veterinary Drug Control (CIVDC), Beijing, China. The other strains were isolated from Qinghai province, China, and stored in our laboratory. *Y. enterocolitica* strains were incubated at 25 °C for 24 h in Luria–Bertani broth (LB) under constant shaking. All other strains were cultured for 18 h in LB broth at 37 °C under constant shaking.

### 2.2. Extraction of Genomic DNA

All bacteria genomic DNA were extracted using the TIANamp Bacteria DNA kit (Tiangen, Beijing, China) following the instructions provided by the manufacturer, and the quantities and qualities were determined by measuring A260 and the A260/A280 ratio with a micro-spectrophotometer (KAIAO, Shanghai, China). All the bacteria genomic DNA were uniformly adjusted to 100 ng/μL and stored at −20 °C until use.

### 2.3. RPA Primer and Probe Designs

In accordance with the reference sequences of *Y. enterocolitica* (accession numbers CP009846), four pairs of primers were designed targeting the conserved region of the *ail* gene. The Exo probe was designed based on this primer amplicon after the primers with high sensitivity were screened by the basal RPA test. The primers and Exo probe were designed following the TwistDx instruction manual (TwistDx, Cambridge, UK). All the primers and the Exo probe were synthesized and provided by Sangon (Sangon Biotech, Shanghai, China) (Table 2).

### 2.4. Real-Time RPA Assay

The real-time RPA reactions were performed in a 50 μL volume using a kit (Amp Future, Weifang, China) according to the manufacturer’s protocol. Briefly, 2.0 μL of 10 μM of both forward and reverse primer, 0.6 μL of 10 μM Probe (RPA-Probe-5), 29.4 μL of A buffer, 10.0 μL DNase-free water, 3.5 μL of the DNA template and 2.5 μL of B buffer were added to a lyophilized RPA reaction pellet. The RPA reactions were run for 30 min at 38 °C, and fluorescence signals were read with a real-time quantitative polymerase chain reaction (Thermo Fisher, Waltham, MA, USA). The RPA-Probe 6 is performed under the same conditions.

### 2.5. Analytical Sensitivity and Specificity

To determine the sensitivity of the RPA assay, the bacterial genomic DNA (100 ng/μL) was diluted in a 10-fold ratio, and six dilution gradients from 10 to 10^−4^ ng/μL were selected for the test. All the samples were tested in duplicate and the whole assay was carried out for three times.

To test the specificity of RPA, DNA was extracted from a group of bacterial chants, described in Table 1, for testing. Three independent reactions were performed.

### 2.6. Repeatability Testing

The bacterial genomic DNA was replicated three times each for inter-batch and intra-batch DNA in three concentration gradients of high (10 ng/μL), medium (10^−1^ ng/μL) and low (10^−5^ ng/μL) concentrations, for a total of nine replicates of the same concentration of DNA.

### 2.7. Validation with Artificially Contaminated Samples

Samples of yak dung were used to assess the potential use and suitability of the real-time RPA assay. All samples were collected from locally farmed yaks and identified as negative for *Y. enterocolitica* by traditional culture assay and PCR. Then, 00 CFU of different *Y. enterocolitica* strains were added to samples without *Y. enterocolitica*. The contaminated samples were added to 5 mL of LB broth and incubated for 24 h at 25 °C to enrich the bacterial concentration. The samples were boiled for 15 min to collect DNA after the enrichment and the RPA reactions were performed.

### 2.8. Data Analysis

Results are expressed as mean ± standard deviation (SD), and all statistical analyses were performed using the SPSS 20.0 software package. The significance of the experimental data was determined by Dunn’s multiple comparisons procedure.

## 3. Results

### 3.1. Optimal Combination Primers and Probes for Real-Time RPA Assay

As the results show, four pairs of primer sets could recognize and amplify *Y. enterocolitica* (Figure 1a,b). It can be seen in Figure 1b that primer F3/R3 has the highest amplification efficiency compared with the three other pairs of primers that can be amplified. Lane 3 in Figure 1a has an indistinct non-specific strip at about 200 bp. In Figure 1a, lane 5 has clearer bands of primer dimers below 100 bp. Therefore, primers F3/R3 with high amplification efficiency and non-specific bands were selected. Comparing the different probes, the probe RPA-Probe-5 amplifies with higher efficiency and sensitivity, as shown in Figure 1c. Therefore, the primer–probe combinations used in this experiment were determined to be RPA-F-3, RPA-R-3 and RPA-Probe-5, respectively. They amplified the target gene fragment size of 192 bp at 38 °C.

### 3.2. Analytical Sensitivity and Specificity of the Real-Time RPA Assay

To determine the sensitivity of the real-time RPA assay, bacterial genomic DNA from 10 to10^−5^ ng/μL were selected for the test. The results showed that the real-time RPA could be detected at a minimum concentration of 10^−4^ ng/μL. The real-time RPA assay could be detected in as little as 3 min (Figure 2a).

The genomic DNA from *Y. enterocolitica* and other bacterial strains was amplified to evaluate the specificity of real-time RPA. The results showed that only pathogenic *Y. enterocolitica* showed significant amplification, with no amplification curve for either non- pathogenic *Y. enterocolitica* or *Y. enterocolitica* (Figure 2b).

This section may be divided by subheadings. It should provide a concise and precise description of the experimental results, their interpretation as well as the experimental conclusions that can be drawn.

### 3.3. Analytical Repeatability of the Real-Time RPA Assay

The bacterial genomic DNA was replicated three times at each of three concentrations: high (10 ng/μL), medium (10^−1^ ng/μL) and low (10^−5^ ng/μL) (Figure 3). Three batches of inter-group replicates were selected to extract DNA separately, using the same kit. The results showed that the high (10 ng/μL), medium (10^−1^ ng/μL) and low (10^−5^ ng/μL) concentrations were stably detected, indicating that the assay has good stability.

### 3.4. Validation of the Real-Time RPA Assay on Artificially Contaminated Samples

To evaluate the performance of the real-time RPA in yak stool samples, the different *Y. enterocolitica* strains were added to different samples. There were five samples identified as *Y. enterocolitica* DNA–positive [Threshold time (Tt) value ranging from 3.17 to 22.47] and five as negative (Tt value undetermined) (Figure 4). Real-time RPA can show significant amplification within 3 min with a clear fluorescent signal for rapid detection of pathogenic *Y. enterocolitica*.

## 4. Discussion

The detection rate of *Y. enterocolitica* has been largely underestimated worldwide because of the difficulties in recovering Yersinia strains from the flora of stool samples [28]. Even though some selective media for Yersinia strains have been developed and isolation of *Y. enterocolitica* has been improved, these selective media are not yet available and therefore cannot be routinely used by clinical laboratories [29,30]. Nowadays, Cepulodin Irgasan Novobiocin Agar (CIN) can be used for the isolation of *Y. enterocolitica*, which contains cefsulodin, irgasen and eosporin as selective antimicrobial agents that inhibit the growth of many members of the *Enterobacteriaceae* family and favor slower growing bacteria [31]. This culture method relies on standard enrichment and selective plating protocols, but this requires additional time, money and work. Here, we developed a real-time RPA method by targeting the *ail* gene for the detection of *Y. enterocolitica*. In our experiment, we selected a sequence specific to the *ail* gene of pathogenic *Y. enterocolitica* for the design of primers and probes. The primers and probes can detect pathogenic *Y. enterocolitis* but not non-pathogenic *Y. enterocolitica*.

The real-time RPA sensitivity experiments can detect a minimum concentration of 10^−4^ ng/μL, which is almost identical to RT-PCR [4] and LAMP [15] sensitivity. It is 10 times more sensitive compared to RPA-SYBR Green I [32] and normal PCR. When the amount of DNA is 35 ng, it can show significant amplification within 3 min with an obvious fluorescence signal, which can rapidly detect *Y. enterocolitica*. The real-time RPA assay in this study did not cross-react with eight other species of pathogens tested, including one species of non-pathogenic *Y. enterocolitica*, or seven clinically common species. Of note, we employed two species of pathogenic *Y. enterocolitica* (*Y. enterocolitica* ATCC 23715 and *Y. enterocolitica* ZDN6) and one non-pathogenic *Y. enterocolitica* species (*Y. enterocolitica* GN 22) to verify the specificity of the method, and the results showed that this method can effectively distinguish between pathogenic *Y. enterocolitica* and non-pathogenic *Y. enterocolitica*.

Existing rapid detection methods (RT-PCR and LAMP) for *Y. enterocolitica* require at least 50 min to complete [4,15]. The detection time of DNA in this study was less than 20 min, and the detection results can be viewed in real time, which greatly improves the detection efficiency of *Y. enterocolitica*. In addition, compared to PCR methods, real-time RPA does not require sophisticated thermal cycling apparatuses and trained personnel. The real-time RPA can detect *Y. enterocolitica* at 38 °C in as little as 3 min. For this study, we used an RT-PCR machine to read the fluorescence signal, but a portable tube scanner can also give satisfactory results [25]. Compared to previous methods, the real-time RPA method is suitable for field testing, and the reagents can be stored at room temperature for long periods of time in the lyophilized format.

The occurrence of pathogenic *Y. enterocolitica* in the conventional environment is not very common, so we applied different *Y. enterocolitica* in order to contaminate samples that tested negative by conventional methods. The real-time RPA reactions can detect different *Y. enterocolitica* species and can accurately distinguish between pathogenic *Y. enterocolitica* and non-pathogenic *Y. enterocolitica*. The real-time RPA reactions are well tolerated for crude samples. In our samples, DNA was released by simple boiling and used directly in the assay without affecting the detection limit or accuracy. This makes the entire assay process simpler. Therefore, the real-time RPA method consists of the two steps of sample boiling and isothermal amplification, and therefore it is easy to operate and has wide application value.

## 5. Conclusions

The present study established a real-time RPA method for the rapid detection of pathogenic *Y. enterocolitica* based on *ail* genes and demonstrated that it could detect pathogenic *Y. enterocolitica* within 20 min. Further study will develop the concept that portable devices based on the current real-time RPA method should be an important tool for early diagnosis of the pathogenic *Y. enterocolitica* outbreak sites in resource-limited settings.

## Figures and Tables

**Figure 1 pathogens-12-00226-f001:**
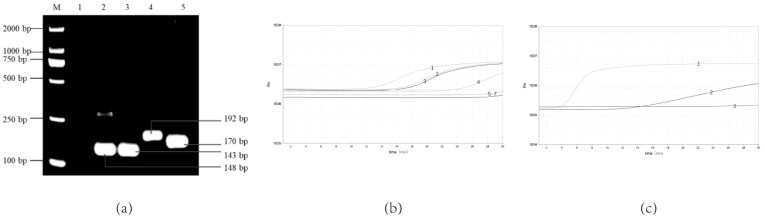
Screening of optimal real-time RPA primer–probe combinations. (**a**) Screening the optimal primers for the real-time RPA by the basic RPA: 1, NTC; 2, F1/R1; 3. F2/R2; 4, F3/R3; 5, F1/R2; (**b**) Screening the optimal primers for the real-time RPA: 1, F3/R3; 2, F1/R1; 3, F2/R2; 4, F1/R2; 5, F1/R3; 6, F2/R3; 7, NTC; (**c**) Screening the optimal probe for the real-time RPA: 1, Probe-5; 2, Probe-6; 3. NCT.

**Figure 2 pathogens-12-00226-f002:**
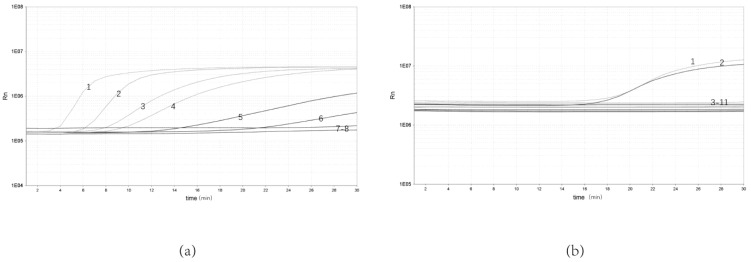
Sensitivity and specificity of real-time RPA assay. (**a**) Sensitivity test of real-time RPA assay: 1, 10 ng/μL; 2, 10^0^ ng/μL; 3, 10^−1^ ng/μL; 4, 10^−2^ ng/μL; 5, 10^−3^ ng/μL; 6, 10^−4^ ng/μL; 7, 10^−5^ ng/μL; 8, NTC. (**b**) Specificity test of real-time RPA assay: 1, *Y. enterocolitica* ATCC 23715; 2, *Y. enterocolitica* ZDN6; 3, *Pasteurella* P0810; 4, *Shigella Castellani* 06-01-03; 5, *Slmonella* 06-03-01; 6, *Escherichia coli* 03-03; 7, *Aeromonas hydrophila AH* 1302; 8, *Bacillus cereus* 06-05-02; 9, non-pathogenic *Y. enterocolitica* GN 22; 10, *Staphylococcus aureus* 04-011; 11, NTC.

**Figure 3 pathogens-12-00226-f003:**
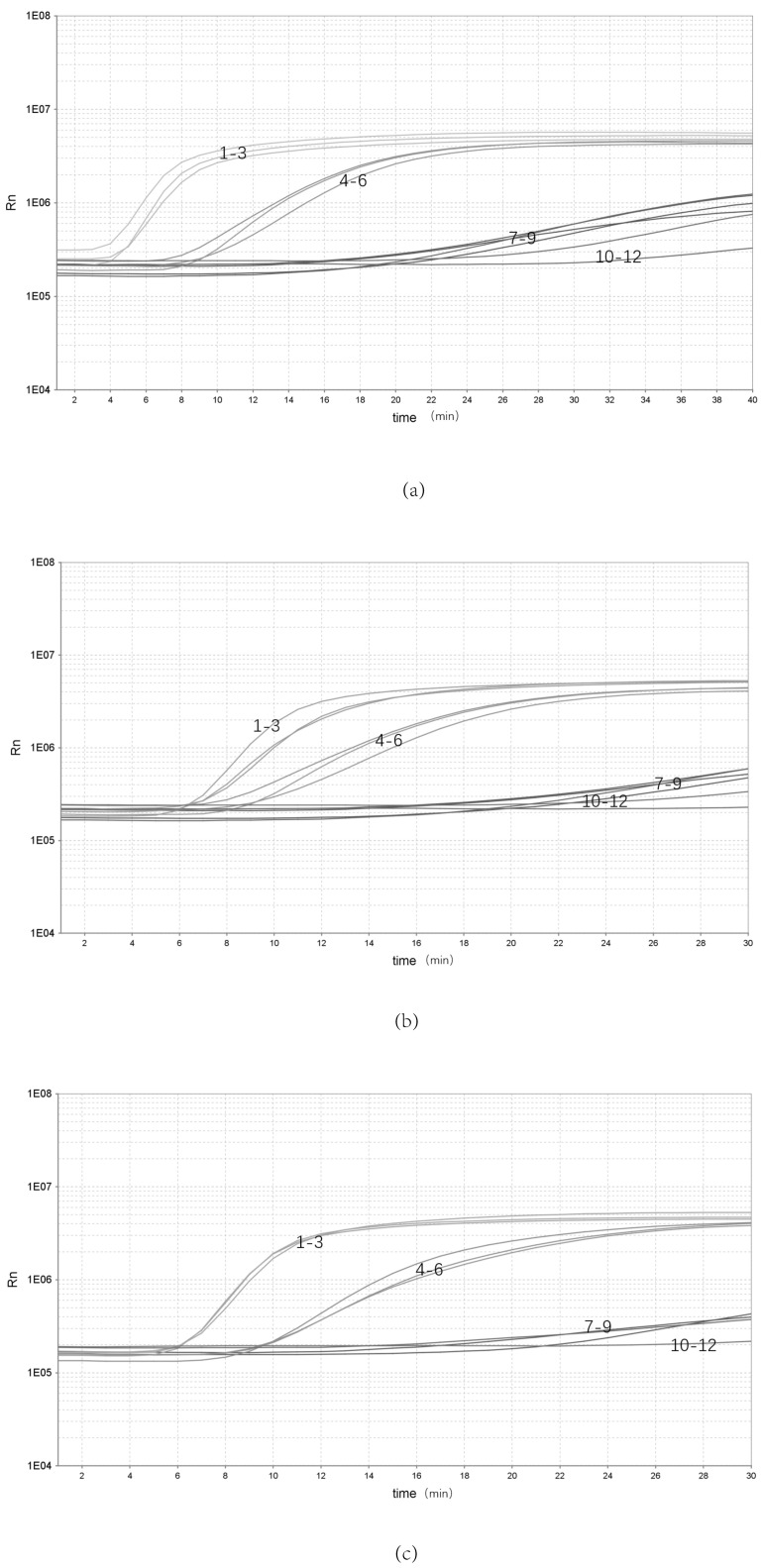
Repeatability of the real-time RPA assay. (**a**–**c**) Repeatability of the real-time RPA assay with different groups: 1–3, 10^0^ ng/μL; 4–6, 10^−2^ ng/μL; 7–9, 10^−5^ ng/μL; 10–12, NTC.

**Figure 4 pathogens-12-00226-f004:**
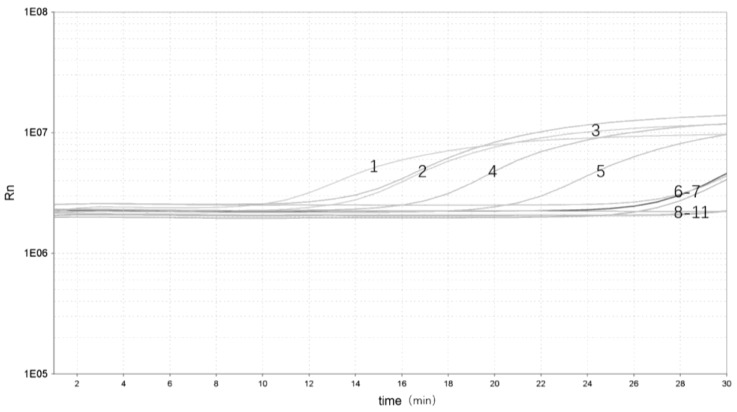
Validation of the real-time RPA assay: 1, pathogenic Y. enterocolitica ZDN 6 (Tt 3.17); 2, pathogenic Y. enterocolitica ATCC 23715 (Tt 9.51); 3, pathogenic Y. enterocolitica 08-01 (Tt 10.02); 4, pathogenic Y. enterocolitica GN 22 (Tt 18.55); 5, pathogenic Y. enterocolitica ZDN 22 (Tt 22.47); 6, non-pathogenic Y. enterocolitica GLN 1 (Tt undetermined); 7, non-pathogenic Y. enterocolitica HNN 18 (Tt undetermined); 8, non-pathogenic Y. enterocolitica HNN 47 (Tt undetermined); 9, non-pathogenic Y. enterocolitica ZDN 37 (Tt undetermined); 10, non-pathogenic Y. enterocolitica HNN 63 (Tt undetermined); 11, NTC (Tt undetermined).

**Table 1 pathogens-12-00226-t001:** All strains used in the study.

Strain	Species	Source	*ail* Gene
ZDN 6	*Y. enterocolitica*	isolated from Yak	+
ATCC 23715	*Y. enterocolitica*	CIVDC	+
08-01	*Y. enterocolitica*	isolated from Landrace	+
GN 22	*Y. enterocolitica*	isolated from Yak	+
ZDN 22	*Y. enterocolitica*	isolated from Yak	+
GLN 1	*Y. enterocolitica*	isolated from Yak	−
HNN 18	*Y. enterocolitica*	isolated from Yak	−
HNN 47	*Y. enterocolitica*	isolated from Yak	−
ZDN 37	*Y. enterocolitica*	isolated from Yak	−
HNN 63	*Y. enterocolitica*	isolated from Yak	−
06-01-03	*Shigella Castellani*	isolated from Yak	−
06-03-01	*Salmonella*	isolated from Feedstuff	−
03-03	*Escherichia coli*	isolated from Feedstuff	−
AH 1302	*Aeromonas hydrophila*	isolated from Gymnocypris przewalskii	−
06-05-02	*Bacillus cereus*	isolated from Feedstuff	−
P0810	*Pasteurella*	isolated from Yak	−
04-011	*Staphylococcus aureus*	isolated from Yak	−

“+” = positive result; “−” = negative result.

**Table 2 pathogens-12-00226-t002:** Real-time RPA primers and probes.

Name	Sequence (5′-3′)
RPA-F-1	TCATGGAAAGGTTAAGGCATCTGTATTTGA
RPA-F-2	ATAGGTTCGTTTGCTTATACTCATCAGGGA
RPA-F-3	AAAGGTTTTAACCTGAAGTACCGTTATGAA
RPA-R-1	TTTTATGCTATCGAGTTTGGAGTATTCATA
RPA-R-2	AGTAATCCATAAAGGCTAACATATTCGTTG
RPA-R-3	TAATCCATAAAGGCTAACATATTCGTTGAT
RPA-Probe-5	AGGTTCGTTTGCTTATACTCATCAGGGATA(FAM-dT) (THF)A(BHQ1-dT) TTCTTCTATGGCAGTA(3′-block)
RPA-Probe-6	GAATAGTAATCAACATCACCATGACCAAACT(FAM-dT) (THF) (BHQ1-dT) TACTGCCATAGAAGA(3′-block)

## Data Availability

The datasets generated during the current study are available from the corresponding author upon reasonable request.

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
