# Peer review of "Establishment of a Real-Time Recombinase Polymerase Amplification for Rapid Detection of Pathogenic Yersinia enterocolitica"

_pathogens, 2023, doi:10.3390/pathogens12020226_

Round 1

Reviewer 1 Report

The manuscript deals with the development of a real-time recombinase polymerase amplification (RPA) method based on the ail gene for detection of pathogenic Yersinia enterocolitica. The method was tested for sensitivity level and specificity, and importantly, using contaminating crude samples (Yak stool samples). The method showed satisfactory performance and reproducibility. The following recommendations to improve the manuscript are listed below.

  1. The sensitivity of the current RT-RPA was established at 10^-4 ng/ul. Please, indicate how much it will be in terms of concentration of bacteria per mL.

  1. Page 1, lines 29-33. Problem with grammar in this sentence.

  1. Page 3, Table 2. Looks like there is a shift in designation of the (3’ block) for the RPA-Probe 5.

  1. Page 3, line 99. If the reaction with RPA-Probe-6 was done at the same condition, this should be indicated here.

  1. Page 4, line 120. “samples” two times

  1. Page 4, lines 121-122. Here it says that DNA was extracted, although in Discussion, p.7, line 223, sample was boiled to release DNA.

  1. Page 4, lines 153-155. Looks like somebody’s note, which is not part of the paper.

Reviewer 2 Report

My comments can be found in the attached WORD file.

Reviewer 3 Report

The article is interesting. The authors present a fast possibility of Yersinia enterocolitica ail gene detection using real-time recombinase polymerase amplification (RPA). Unfortunately, ail gene is also in other species of Yersinia, including Y. pseudotuberculosis or Y. pestis. The authors must perform studies with other Yersinia species, to eliminate cross-detection because the primers used can also react with other Yersinia species.

Round 2

Reviewer 2 Report

The authors addressed most of my questions in their response to the reviewer. However, they did not address these questions in their manuscripts. For example, they do not discuss anything about Tt values in their manuscript, even though these values were used as a basis to decide whether a test is considered positive or not. 

I suggest that the authors' responses to Points 2, 5, 8 and 9 be incorporated into the result or discussion sections of their manuscript to better explain the experimental set-ups and their logics for the result interpretations, not just to me but to the potential readers.

Round 3

Reviewer 2 Report

I thank the authors have sufficiently addressed my previous concerns and I thank them for their time and effort.